# The Lectin-Like Domain of Thrombomodulin Inhibits β1 Integrin-Dependent Binding of Human Breast Cancer-Derived Cell Lines to Fibronectin

**DOI:** 10.3390/biomedicines9020162

**Published:** 2021-02-07

**Authors:** Eiji Kawamoto, Nodoka Nago, Takayuki Okamoto, Arong Gaowa, Asami Masui-Ito, Yuichi Akama, Samuel Darkwah, Michael Gyasi Appiah, Phyoe Kyawe Myint, Gideon Obeng, Atsushi Ito, Siqingaowa Caidengbate, Ryo Esumi, Takanori Yamaguchi, Eun Jeong Park, Hiroshi Imai, Motomu Shimaoka

**Affiliations:** 1Department of Molecular Pathobiology and Cell Adhesion Biology, Mie University Graduate School of Medicine, 2-174 Edobashi, Tsu-city, Mie 514-8507, Japan; arong-g@doc.medic.mie-u.ac.jp (A.G.); shironeko1am@yahoo.co.jp (A.M.-I.); y-akama@nms.ac.jp (Y.A.); kwekuadarkwah@gmail.com (S.D.); 317ds06@m.mie-u.ac.jp (M.G.A.); 317ds09@m.mie-u.ac.jp (P.K.M.); 318ds16@m.mie-u.ac.jp (G.O.); i-atsushi@clin.medic.mie-u.ac.jp (A.I.); 320d011@m.mie-u.ac.jp (S.C.); ryoo1582@hotmail.co.jp (R.E.); norinori_0712@yahoo.co.jp (T.Y.); epark@doc.medic.mie-u.ac.jp (E.J.P.); motomushimaoka@gmail.com (M.S.); 2Department of Emergency and Disaster Medicine, Mie University Graduate School of Medicine, 2-174 Edobashi, Tsu-city, Mie 514-8507, Japan; hi119@clin.medic.mie-u.ac.jp; 3Department of Clinical Nutrition, Suzuka University of Medical Science, 1001-1 Kishioka-cho, Suzuka-city, Mie 510-0293, Japan; nagou@suzuka-u.ac.jp; 4Department of Pharmacology, Faculty of Medicine, Shimane University, 89-1 Enya-cho, Izumo-city, Shimane 693-8501, Japan; okamoto@med.shimane-u.ac.jp

**Keywords:** thrombomodulin, integrin, breast cancer cell, cell adhesion, V-well assay

## Abstract

Thrombomodulin is a molecule with anti-coagulant and anti-inflammatory properties. Recently, thrombomodulin was reported to be able to bind extracellular matrix proteins, such as fibronectin and collagen; however, whether thrombomodulin regulates the binding of human breast cancer-derived cell lines to the extracellular matrix remains unknown. To investigate this, we created an extracellular domain of thrombomodulin, TMD123-Fc, or domain deletion TM-Fc proteins (TM domain 12-Fc, TM domain 23-Fc) and examined their bindings to fibronectin in vitro by ELISA. The lectin-like domain of thrombomodulin was found to be essential for the binding of the extracellular domain of thrombomodulin to fibronectin. Using a V-well cell adhesion assay or flow cytometry analysis with fluorescent beads, we found that both TMD123-Fc and TMD12-Fc inhibited the binding between β1 integrin of human breast cancer-derived cell lines and fibronectin. Furthermore, TMD123-Fc and TMD12-Fc inhibited the binding of activated integrins to fibronectin under shear stress in the presence of Ca^2+^ and Mg^2+^ but not under strong integrin-activation conditions in the presence of Mg^2+^ without Ca^2+^. This suggests that thrombomodulin Fc fusion protein administered exogenously at a relatively early stage of inflammation may be applied to the development of new therapies that inhibit the binding of β1 integrin of breast cancer cell lines to fibronectin.

## 1. Introduction

Thrombomodulin (TM) is considered an important cofactor of the anticoagulation protein C system present in vascular endothelial cells [1]. In addition, TM can exert anti-inflammatory and cytoprotective properties and was found to be a key component regulating the binding of vascular endothelial cells to white blood cells [2]. TM is expressed in tumor cells in addition to normal vascular endothelial cells, with its expression in tumor cells suggested to regulate tumor cell invasion and proliferation in certain types of cancer. For example, in esophageal cancer, TM expression in cancer cells in metastases is reduced relative to that in primary tumors [3]. Furthermore, low TM expression in invasive breast cancer cells is significantly correlated with cancer recurrence rates [4]. These findings suggest that TM might play an important role in cancer growth and metastasis. A recent study showed that TM binds to the extracellular matrix (ECM), including fibronectin (FN), collagen, and laminin, of vascular endothelial cells and tumor cells [5]. However, it remains unknown whether TM regulates the binding of human breast cancer-derived cell lines to the ECM.

Integrins are composed of α- and β-subunit heterodimers, with 18 α and 8 β subunits and 24 α and β heterodimer combinations identified [6]. Among these integrins, β1 integrin is expressed by almost all cells and binds to ECM components, such as fibronectin, collagen, and laminin. It also connects the actin filament-like cytoskeletal system within the cell and plays an important role in cell adhesion. In particular, α5β1 integrin plays an important role in cell adhesion, migration, proliferation, and differentiation in both normal and tumor cells [7]. For example, the association between elevated expression levels of β1 integrin in cancer cells and poor prognosis has been reported in tumor cells such as breast [8,9] and lung cancer [10]. In a breast cancer model using β1 conditional knockout (KO) mice, the involvement of β1 integrins on the surface of cancer cells has been emphasized in breast cancer development [11] and metastasis [12]. Therefore, studies on the molecular interactions between integrin α5β1 and its ligands might be essential to interpret the biological functions and underlying mechanisms of α5β1 integrin.

We hypothesized that if TM inhibits the binding of β1 integrin from human breast cancer-derived cell lines to fibronectin in the ECM, then it might represent a new therapeutic strategy for inhibiting cancer growth and metastasis. To test this hypothesis, in this study, we generated TMD123-Fc and domain-deletion TM proteins (TMD12-Fc and TMD23-Fc) fused with an extracellular domain of TM (TMD123) and human IgG Fc fusion protein. These Fc fusion proteins contain extracellular domains expressed on the luminal side of the vessel among the TM components and might contact blood cells or tumor cells suspended in the vessel or lymph node. We analyzed these fusion proteins for their effects on the binding of β1 integrin from human breast cancer-derived cell lines (i.e., MDA-MB-231 and MCF-7) to fibronectin. Our results will contribute to the development of new therapeutic strategies to inhibit the metastasis of breast cancer cells.

## 2. Materials and Methods

### 2.1. Construction of a Recombinant Human TM (Domains 1, 2, and 3)-Immunoglobulin Fc Fusion Protein (TMD123-Fc) Expression Vector

A DNA fragment containing TM domains 1, 2, and 3 was amplified by PCR from the pSV2TMJ2 vector [13], which contains the entire sequence of human TM. The TMD123 sequence was subcloned into the human IgG1 Fc frame of the pcDNA3.1(+) vector at the *Hin*dIII/*Bam*HI site [14]. Plasmids for domain deletion mutants of soluble TM-Fc fusion proteins (i.e., TM domain 23-Fc lacking domain 1, TM domain 12-Fc lacking domain 3, and Fc lacking domains 1, 2, and 3) were generated by inverse PCR using the TM domain 123-Fc-containing pcDNA3.1(+) plasmid as the template.

### 2.2. Expression and Purification of TM Domain-Fc Fusion Proteins

TM domain 123-Fc or mutant TM-Fc plasmids (TM domain 12-Fc, TM domain 23-Fc, and Fc) were transiently transfected into the human embryonic kidney cell line 293T (HEK293T) cells (ATCC no. CRL-1573; ATCC, Manassas, VA, USA) using Lipofectamine 2000 reagent (Invitrogen, Tokyo, Japan). HEK293T cells were cultured in Opti-MEM (Gibco; Invitrogen, Tokyo, Japan) without fetal bovine serum (FBS) for 7 days at 37 °C, and the supernatant was collected. TM domain 123-Fc and other mutant proteins secreted in the supernatant were purified with protein A-affinity using the Amicon Pro purification system (Millipore Japan, Tokyo, Japan). The purity of TM domain 123-Fc and mutant proteins was confirmed by SDS-PAGE [14].

### 2.3. V-Well Cell Adhesion Assay

A cell adhesion assay using a 96-well V-well plate was performed as previously described [15]. The V-well was coated with 100 μL of 10 µg/mL fibronectin (Sigma-Aldrich, Tokyo, Japan) or control 1% bovine serum albumin (BSA) as an integrin ligand. Plates were incubated overnight at 4 °C and blocked with phosphate-buffered saline (PBS) containing 1% BSA for 2 h at 37 °C. A solution of 100 μL of HEPES-buffered saline (HBS) containing 1 × 10^4^ calcein-labeled MDA-MB-231 β1 integrin wild type (WT), MDA-MB-231 β1 integrin-KO cells, or MCF-7 cells in either 1 mM Mg^2+^/Ca^2+^ or 2mM EDTA was dispensed into each well. TMD123-Fc, mutant TM-Fc fusion protein, or control human IgG1 Fc protein were dissolved in PBS and aliquoted into each well to confirm the inhibitory effect of fibronectin and β1 integrin-mediated cell adhesion. Conversion of molar concentrations of proteins was as follows: 0.25 µM TMD123-Fc to 38.8 µg/mL, 0.25 µM TMD12-FC to 37.2 µg/mL, 0.25 µM TMD23-Fc to 27.1 µg/mL, and 0.25 µM Fc to 12.8 µg/mL. Before centrifugation, V-wells with cells were incubated at 37 °C for 5 min. Shear stress was loaded to confirm integrin-dependent binding. To load the shear stress, the plates were centrifuged at 280 g (1200 rpm) for 5 min using a swing-bucket rotor (EX-125; Takaratomy Seiko Co., Ltd., Tokyo, Japan). Non-adherent cells that accumulated at the bottom of the wells were detected using the 2030 ARVO X-2 Multilabel Reader (PerkinElmer Japan, Kanagawa, Japan). We calculated the binding affinity between cells and integrin ligands, as follows:Binding Affinity (%) = {FL(EDTA) − FL(Mg^2+^/Ca^2+^)}/FL(EDTA) × 100
where FL(EDTA) represents the fluorescence intensity of integrin binding to the integrin ligand in the presence of 2 mM EDTA, and FL(Mg^2+^/Ca^2+^) represents the fluorescence intensity of integrin binding to the integrin ligand in the presence of 1 mM Mg^2+^/Ca^2+^. In the presence of 2 mM EDTA, non-adherent cells were concentrated at the bottom of the V-well to elicit an increase in fluorescence intensity. In the presence of 1 mM Mg^2+^/Ca^2+^, integrins were activated, and cells adhered to integrin ligands attached to the V-well, resulting in decreases in fluorescence intensity. Introduction of an adhesion inhibitor, such as an integrin antibody or TM, resulted in aggregation of non-adherent cells at the bottom of the V-well and increased fluorescence intensity.

### 2.4. Cell Culture

The HEK293T, MDA-MB-231, and MCF-7 cell lines were maintained at 37 °C in Dulbecco’s modified Eagle medium supplemented with 10% FBS in a humidified atmosphere containing 5% CO_2_. All cell lines were purchased from ATCC.

### 2.5. β1. Integrin KO Cell with CRISPR/CAS9 System

The sgRNA-specified oligo sequence of human β1 integrin exon 2 (forward; 5′-CACCGGAGGAATGTTACACACGGCTGC-3′, reverse; 5′-AAACGCAGCCGTGTAACATTCCTCCTCCTCC-3′) was cloned into the pSpCas9(BB)-2A-Puro (PX459) V2.0 (Addgene plasmid ID: 62988; Addgene, Watertown, MA, USA) vector [16]. pSpCas9(BB)-2A-Puro (PX459) V2.0 was a gift from Feng Zhang (plasmid #62988; Addgene). This plasmid was transfected into MDA-MB-231 cells using the Lipofectamine 2000 reagent (Invitrogen). After 72 h of transfection, the cells were selected with 10 μg/mL of puromycin. After 10 days of incubation, colony-forming β1 integrin KO cells were picked up and incubated further in a 96-well plate and selected twice. β1 integrin KO cells were identified by flow cytometric analysis.

### 2.6. Enzyme-Linked Immunosorbent Assay (ELISA)

TMD123-Fc and its domain mutant protein were immobilized in 96-well flat plates at 4 °C overnight. The immobilized protein concentration is as follows: 0.25 µM TMD123-Fc to 38.8 µg/mL, 0.25 µM TMD12-FC to 37.2 µg/mL, 0.25 µM TMD23-Fc to 27.1 µg/mL, and 0.25 µM Fc to 12.8 µg/mL. The next day, the wells were washed twice with PBS, 100 μL of 1 μg/mL FN was injected into the wells, and the plate was allowed to incubate for 1 h at room temperature. The wells were then washed twice with PBS and incubated with 10 µg/mL of anti-human fibronectin antibody (HRP-conjugated detection antibody; cat. no. 1470-05; SouthernBiotech, Birmingham, AL, USA) for 1 h at 4 °C. Thereafter, the wells were washed twice with PBS. For the detection of fibronectin binding to TMD123-Fc, the OD450 was measured according to the instructions of the ELISA kit (BD OptEIA; cat. no. 550534; BD Biosciences, Franklin Lakes, NJ, USA).

### 2.7. Flow Cytometry Analysis

MDA-MB-231 and MCF-7 cells were grown to 90% confluence, detached from the culture dish using trypsin with 1 mM EDTA, washed with FACS buffer (PBS with 2 mM EDTA, 2% BSA, and 0.05% NaN3), and stained with the β1 integrin primary antibody (10 µg/mL) for 30 min at room temperature. The sample was then incubated with a secondary antibody (FITC-anti-mouse IgG, 15 µg/mL) for 30 min at room temperature. The cells were washed three times with FACS buffer and analyzed for integrin expression using a BD Accuri C6 system (BD Biosciences).

### 2.8. Antibodies Targeting Integrins

The following antibodies were used for the identification of integrins: anti-integrin β1 monoclonal antibody (mAb; clone P5D2, MAB17781; R&D Systems, Minneapolis, MN, USA), β1 mAb (clone P5D2, sc-13590, lot K2618; Santa Cruz Biotechnology, Dallas, TX, USA), β3 mAb (clone VIPL2, ab92393, lot GR274279-1; Abcam, Cambridge, UK), β6 mAb (clone CSbwta6, NBP2-29800, lot 3120164; NOVUS, Littleton, CO, USA), α6 mAb (clone BQ16, sc-13542; Santa Cruz Biotechnology), and mouse IgG1 isotype control antibody (clone MOPC-21, 70-4714, lot P4714032717703; Tonbo, San Diego, CA, USA). FITC-conjugated goat anti-mouse IgG (H+L, cat. no. 115-095-003; Jackson ImmunoResearch Laboratories, West Grove, PA, USA) was used as a secondary antibody. Other antibodies included the following: anti-integrin α1 mAb-PE (clone TS2/7, cat. no. 328304, Lot B268325; Biolegend, San Diego, CA, USA), anti-integrin α2 mAb-PE (clone P1E6-C5 PE, cat. no. 359308; Biolegend), anti-integrin α3 mAb-PE (clone ASC-1, cat. no. 343803, lot B266733; Biolegend), anti-integrin α4 mAb-PE (clone 9F10, cat. no. 304304, lot B288306; Biolegend), anti-integrin α5 mAb-PE (clone NK1-SAM-1, cat. no. 328010, lot B268740; Biolegend), anti-integrin αV mAb-FITC (clone P2W7, cat. no. 134480, lot 100959; Lifespan Biosciences, Seattle, WA, USA), anti-integrin α8 mAb-PE (clone 481709, cat. no. MA5-23677, UK2877873; Invitrogen), anti-integrin αIIb (CD41) mAb-FITC (clone H1P8, cat. no. 303704, lot B239666; Biolegend), anti-integrin β5 mAb-PE (clone AST-3T (cat. no. 345203, lot B283165; Biolegend), integrin β7 mAb-PE (clone FIB504, cat. no. 321204, lot B274285; Biolegend), mouse IgG1 isotype control antibody-PE (clone MOPC-21, cat. no. 400112, lot B273446; Biolegend), mouse IgG1 isotype control antibody-FITC (clone MOPC-21, cat. no. 400108, lot B258679; Biolegend), mouse IgG2b isotype control antibody-PE (clone MPC-11, cat. no. 400312, lot B285393; Biolegend), and rat IgG2a isotype control antibody-PE (clone RTK2758, cat. no. 400508, lot B316163; Biolegend).

### 2.9. Cell Adhesion Experiments Using Fibronectin-Coated Fluorescent Beads

Fluorescent beads (20 µL; 3 µm in diameter; cat. no. 17155; Fluoresbrite Plain Microspheres; Polysciences, Inc., Warrington, PA, USA) were mixed with 10 µg fibronectin or 10 µg BSA and incubated at room temperature for 15 min. PBS (1 mL) was then added to the tube, and the beads were incubated at 4 °C for 12 h. The following day, the beads were blocked with 1 M glycine for 30 min at room temperature and washed twice with 1% BSA/PBS. The beads were dissolved in 100 µL of 1% BSA/PBS, and 5 µL was mixed with 6 × 10^4^ MDA-231 cells and simultaneously mixed with either 2 mM EDTA, 1 mM Mg^2+^/Ca^2+^, 1 mM Mg^2+^, 0.25 µM TM-Fc proteins, or 10 µg/mL β1 integrin antibody and incubated on a flat shaker at 150 rpm (cat. no. SK-O180-E; DLAB, Co., Ltd., Beijing, China) for 15 min at room temperature. The binding of fibronectin-coated fluorescent beads to MDA-231 was measured by flow cytometry.

### 2.10. Statistical Analysis

Statistical analyses were performed using SPSS software (v.26.0; IBM Corp., Armonk, NY, USA). Results are presented as the mean ± standard deviation. Wilcoxon and Mann–Whitney tests were used for within-group comparisons. Statistical significance was set at *p* < 0.05.

## 3. Result

### 3.1. The Lectin-Like Extracellular Domain of TM Binds Fibronectin

To elucidate which of the three extracellular domains (lectin-like domain (domain 1; D1), epidermal growth factor (EGF)-like domain (domain 2; D2), and serine-threonine-rich domain (domain 3; D3)) of TM binds fibronectin, we fused the extracellular domain with human IgG Fc protein to create the TMD123-Fc protein (Figure 1). Two different concentrations (0.25 µM, 0.1 µM) of TMD123-F, TMD12-Fc, TMD23-Fc, or Fc were immobilized in the plate. Subsequently, we observed that fibronectin bound to the TM protein containing the lectin-like domain (*p* < 0.05) (Figure 2), whereas TMD23-Fc did not bind fibronectin. This result suggested that the lectin-like domain of TM is required for the binding of TMD123-Fc to fibronectin.

### 3.2. Binding of Human Breast Cancer-Derived Cell Lines to Fibronectin Is Inhibited by TMD123-Fc in a Concentration-Dependent Manner

Fibronectin is the major ligand for β1 integrin [17]. A previous study demonstrated that αIIbβ3, αVβ3, αVβ6, αVβ1, α5β1, α8β1, α4β1, and α4β7 integrins bind fibronectin [18]. Among these integrins, α5β1 integrin is expressed on MDA-MB-231 and MCF-7 cells (Figure 3 and Appendix A
Appendix A). We then generated β1 integrin-KO MDA-MB-231 cells using the CRISPR/Cas9 β1 integrin-KO system (Figure 4). Surprisingly, the α5 subunit expressed on the cell surface was also lost. This suggests that the integrins must have a coordinated presence of αβ subunits on the cell surface. Next, to investigate whether MDA-MB-231 was inhibited by TMD123-Fc through the binding of β1 integrin to fibronectin, we evaluated binding of the β1 integrin to fibronectin by adding a divalent ion (1 mM Mg^2+^ and Ca^2+^) to immobilized fibronectin and activating the integrin by applying shear stress to the β1 integrin from human breast cancer-derived cell lines. We found that the binding of fibronectin to β1 integrin was inhibited by TMD123-Fc in a concentration-dependent manner (*p* < 0.05) (Figure 5a,c).

### 3.3. The Lectin-Like Domain of TM Inhibits the Binding of β1 Integrin from Human Breast Cancer-Derived Cell Lines to Fibronectin

Among the extracellular domains of TM, the lectin-like domain binds fibronectin (Figure 2). Therefore, we hypothesized that the binding of human breast cancer-derived cell lines to fibronectin would be inhibited by TMD123-Fc (0.25 µM) and TMD12-Fc (0.25 µM) but not by TMD23-Fc (0.25 µM). To test this hypothesis, fibronectin (10 μg/mL) was immobilized in V-well plates and MDA-MB-231 or MCF-7 β1 integrin was activated by shear stress in the presence of 1 mM Mg^2+^ and Ca^2+^. Only TMD123-Fc and TMD12-Fc inhibited the binding of fibronectin to integrin, indicating that only proteins with the TM lectin-like domain inhibited binding (*p* < 0.05) (Figure 5b,d). This result suggests that the lectin-like domain of TM plays an important role in inhibiting fibronectin binding to β1 integrin.

### 3.4. TM Inhibits the Integrin-Dependent Binding of MDA-MB-231 to Fibronectin in the Presence of Ca^2+^

There is no Ca^2+^-binding site in the TM lectin-like domain [19]; however, the EGF-like extracellular domain of TM exerts its ability to activate protein C by binding Ca^2+^ [20,21]. We confirmed the binding of fibronectin to human breast cancer-derived cell lines under two conditions (in the presence of Mg^2+^/Ca^2+^ or Mg^2+^) in order to examine whether the presence or absence of Ca^2+^ is important for TM binding to fibronectin. We found that binding of MDA-MB-231 to FN was inhibited in the presence of Mg^2+^ and Ca^2+^ (*p* < 0.05) but not in the presence of Mg^2+^ (Figure 6).

### 3.5. Binding of MDA-231 Cells to Fibronectin Is Inhibited by TMD123-Fc

We then examined whether TMD123-Fc could inhibit the binding of MDA-231 to fibronectin using flow cytometry analysis. Fibronectin or BSA was adsorbed onto the surface of fluorescent beads (green; diameter: 3 μm), and binding of MDA-231 to the beads was confirmed by flow cytometry in the presence of 1 mM Mg^2+^/Ca^2+^, 1 mM Mg^2+^, and 2 mM EDTA (Figure 7a). We found that fibronectin-coated beads bound MDA-231 β1 integrin in the presence of 1 mM Mg^2+^/Ca^2+^ and 1 mM Mg^2+^ but not 2 mM EDTA. Furthermore, the binding of fibronectin-coated beads to MDA-231 cells was inhibited by TMD123-Fc and TMD12-Fc (Figure 7b).

## 4. Discussion

The integrin family of cell-adhesion receptors regulates a diverse range of cellular functions important for adhesion, migration, and metastasis of solid tumors. Integrin α5β1 recognizes and attaches to extracellular ligands containing RGD tripeptide motifs. Aberrant upregulation of integrin α5β1 has been implicated in many human malignancies and is closely associated with poor prognosis [22]. Therefore, studies on the molecular interactions between integrin β1 and its ligands are essential for interpreting the biological functions and underlying mechanisms associated with β1 integrin. The lectin-like domain of TM binds the ECM protein fibronectin; however, it remains unknown whether TM inhibits the adhesion of fibronectin to cancer cell integrins [5]. Here, we created recombinant proteins of the extracellular domain of TM and found that TMD123-Fc and TMD12-Fc inhibited the β1 integrin-dependent binding of human breast cancer-derived cell lines to fibronectin (Figure 5, Figure 6 and Figure 7). Hence, TMD123-Fc administered into the bloodstream may inhibit fibronectin-mediated adhesion of breast cancer cells to the vascular endothelial cells and suppress cancer metastasis.

TM binds fibronectin (Figure 2) and is intimately involved in breast cancer invasion and metastasis [4]. TM is present in both the cytoplasm and on the cell surface of breast cancer cells, as well as in endothelial cells proximal to or in cancerous tissue. TM expression on the surface of breast cancer cells is significantly correlated with a high recurrence rate. Furthermore, TM expressed on the surface of tumor cells is involved in tumor invasion and angiogenesis via binding to the ECM [5]. Therefore, sufficient exogenously administered doses of TM might bind fibronectin on the vascular endothelium and inhibit the binding of tumor cell β1 integrin to fibronectin.

As a way to increase the blood concentration of TMD123, which is certified for the treatment of septic disseminated intravascular coagulation (DIC), and as a way to enhance the drug’s effect on rhsTM (also known as Recomodulin, which is an approved drug for DIC treatment), we focused on the Fc fusion protein, which has been used in several successful biopharmaceuticals in the past few decades. [23]. Based on the pharmacokinetic data obtained from healthy volunteers and patients with DIC, a once-daily intravenous infusion of 0.06 mg/kg rhsTM was recommended [24]. The half-life of rhsTM is approximately 20 h, and more than 50% of the drug is excreted in urine. Our TMD123-Fc fusion protein not only showed the bioactivity of TM itself, but also displayed a long blood half-life conferred by the Fc region and is expected to maintain long-term effects owing to the slower renal clearance of larger sized molecules [25]. Therefore, TMD123-Fc may be a promising candidate for future therapeutic agents along with rhsTM.

We examined the inhibitory effect of TM on the binding of α5β1 integrin to fibronectin using human breast cancer-derived cell lines. Fibronectin also binds to other integrins [18]. MDA-MB-231 and MCF-7 do not harbor α4β1 integrin; however, neutrophils bind fibronectin via α4β1 integrin under shear stress [26]. Therefore, neutrophil α4β1 integrin might bind to fibronectin present on the surface of vascular endothelial cells, and TMD123-Fc might inhibit this adhesion. Thus, in diseases such as sepsis where inhibition of excessive adhesion of neutrophils and vascular endothelial cells is important for the treatment, TMD123-Fc may be a promising candidate for a therapeutic agent. Indeed, exogenously administered rhsTM inhibits leukocyte and vascular endothelial cell adhesion both in vitro and in vivo [27,28]. Therefore, the inhibitory effect of TMD123-Fc with β1 integrin and fibronectin could be applied as a therapeutic strategy for sepsis.

Approximately 1 mM of Ca^2+^ and Mg^2+^ are present in the blood, and the presence of Ca^2+^ plays an important role in maintaining integrins in an inactive state. In our experiments, we activated integrins using shear stress in the presence of 1 mM of Ca^2+^ and Mg^2+^ to mimic physiological blood conditions [29,30]. Notably, the removal of Ca^2+^ or the addition of Mg^2+^ significantly increases the ligand binding affinity of the integrins to ligands. This situation mimics the so-called excessive inflammation state. Our results showed that the absence of Ca^2+^ forced integrin activation and under these conditions, the binding of MDA-MB-231 and MCF-7 to fibronectin was not inhibited by TMD123-Fc administration (Figure 6). This suggests that once integrin is activated, TMD123-Fc is unable to inhibit the binding of MDA-MB-231 β1 integrin to fibronectin. Therefore, it may be essential to administer TMD123-Fc in the early stages of inflammation, before integrins are activated. This is consistent with the clinical results that early administration of rhsTM is more effective in the treatment of sepsis [31].

There are several limitations to this study. First, β1 integrin-KO animals usually die within a few hours after birth likely due to dehydration and loss of the required epidermal barrier [32]. The removal of β1 integrin puts high stress on the cell; therefore, loss of β1 integrin might adversely affect other integrin functions, such as proliferation and signal transduction, in addition to cell adhesion. Thus, adhesion of β1 integrin-KO cells to fibronectin might be reduced beyond β1-dependent binding (Figure 5a,b). Second, we found that TMD123-Fc inhibited the binding of human breast cancer-derived cell lines to fibronectin in vitro. However, we have not conducted in vivo experiments. A longer half-life may be necessary to prove the efficacy of TMD123-Fc in vivo. Third, in the current study, we did not include the rhsTM available drug for the anticoagulant therapy for sepsis-induced disseminated intravascular coagulation (DIC). We found that TMD123-Fc inhibits the binding of β1 integrin of human breast cancer-derived cell lines to fibronectin. Therefore, it has also been suggested that the rhsTM inhibits the binding of β1 integrin of human breast cancer-derived cell lines to fibronectin, but we have not demonstrated whether rhsTM has the same effect as TMD123-Fc in the current study. If future studies prove that rhsTM has similar effects, rhsTM could be rapidly introduced as a new therapeutic approach for the treatment of cancer. Finally, breast cancer is most commonly spread via lymphatics [33]. However, breast cancer migrates to distant organs by lymphogenous and hematogenous metastasis [34,35]. Because fibronectin is present in vascular endothelial cells, lymphatic vessels [36], and lymph nodes [37], TMD123-Fc might contribute to the inhibition of β1 integrin binding to fibronectin at both vascular endothelial cells and lymph nodes.

## 5. Conclusions

We found that both TMD123-Fc and TMD12-Fc, which included the lectin-like domain of TM, inhibited the binding of shear stress-activated β1 integrin from human breast cancer-derived cell lines to fibronectin. These results could be applied to develop new therapeutic strategies for inhibiting the adhesion of breast cancer cells to vascular endothelial cells through β1 integrin-mediated cell adhesion.

## Figures and Tables

**Figure 1 biomedicines-09-00162-f001:**
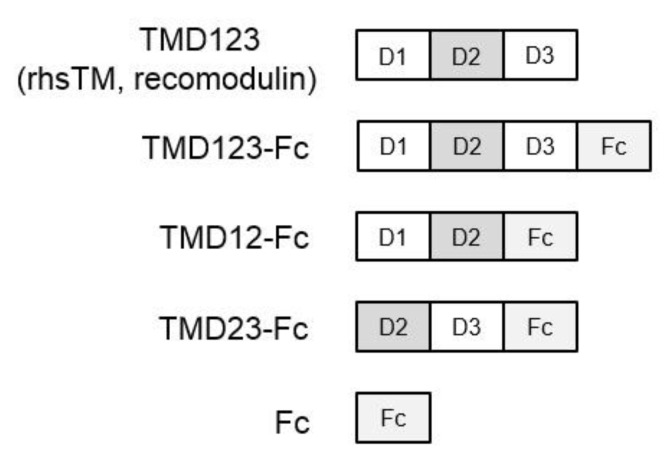
Illustration of extracellular domains of Thrombomodulin (TM) and domain-deleted mutant TM-Fc proteins. TMD123 is also known as rhsTM (Recomodulin), a treatment drug for septic disseminated intravascular coagulation. TM has large extracellular domains (domains 1–3): N-terminal C-type lectin-like domain (D1), the epidermal growth factor-like domain (D2), and the membrane proximal serine/threonine-rich domain (D3). Fc represents the human IgG Fc region.

**Figure 2 biomedicines-09-00162-f002:**
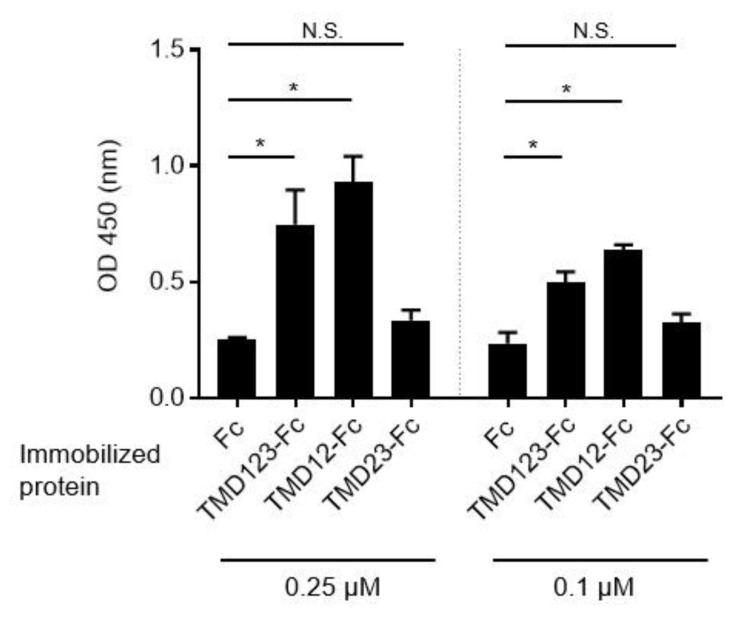
Adhesion of extracellular domains of TM (TM) and domain-deleted mutant TM-Fc proteins to fibronectin (FN). Two concentrations of TM-Fc were immobilized in 96-well flat plates, and after FN (1 µg/mL) was administered, adhesion was detected by ELISA using anti-fibronectin-HRP. Experiments were performed in triplicate and repeated three times with similar results. OD (optical density) was read at 450 nm. Data are expressed as mean ± SD. * *p* < 0.05 compared with Fc. N.S.: not significant.

**Figure 3 biomedicines-09-00162-f003:**
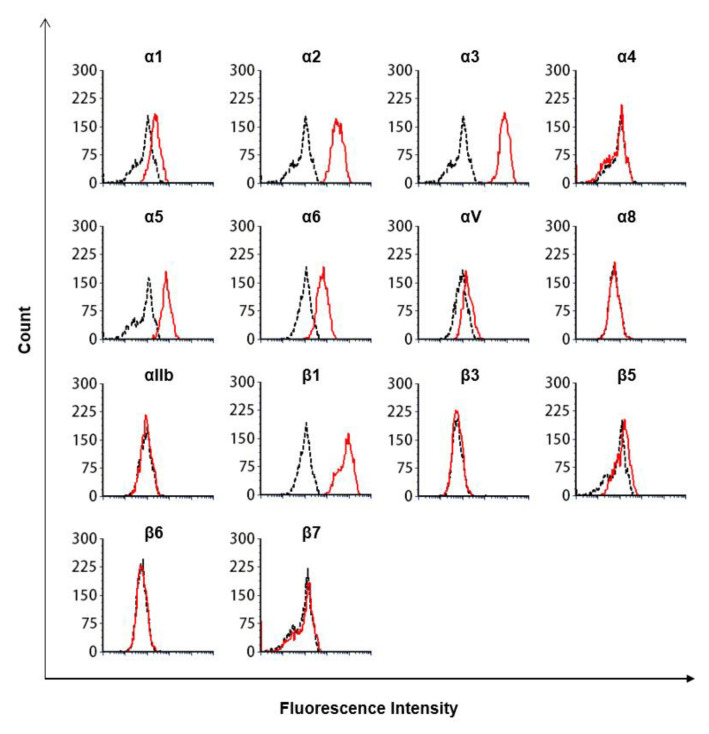
Flow cytometry analysis of integrin expression on MDA-MB-231 cells. Dotted lines represent isotype controls, and solid lines represent integrin expression. Results are representative of three experiments with similar results.

**Figure 4 biomedicines-09-00162-f004:**
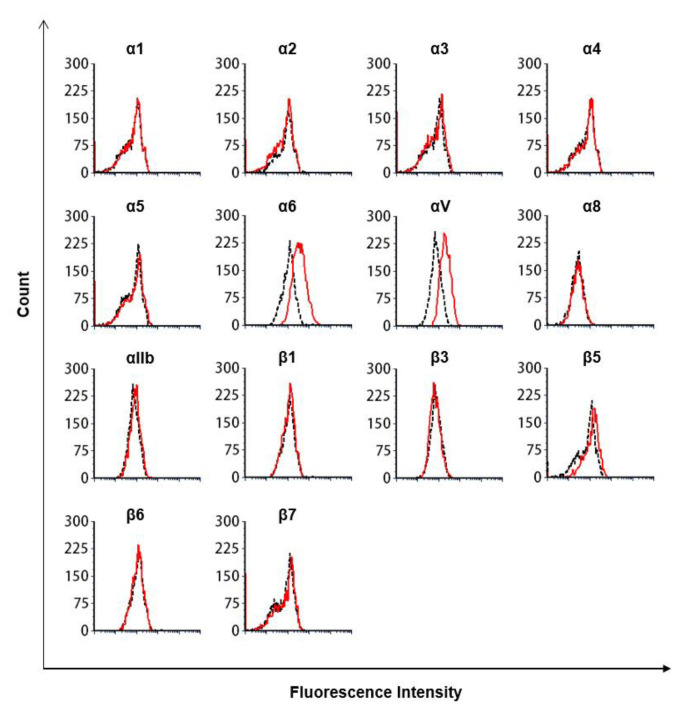
Flow cytometry analysis of integrin expression on MDA-MB-231 β1 integrin-KO cells. Dotted lines represent isotype controls, and solid lines represent integrin expression. The figure is a representative of three experiments with similar results.

**Figure 5 biomedicines-09-00162-f005:**
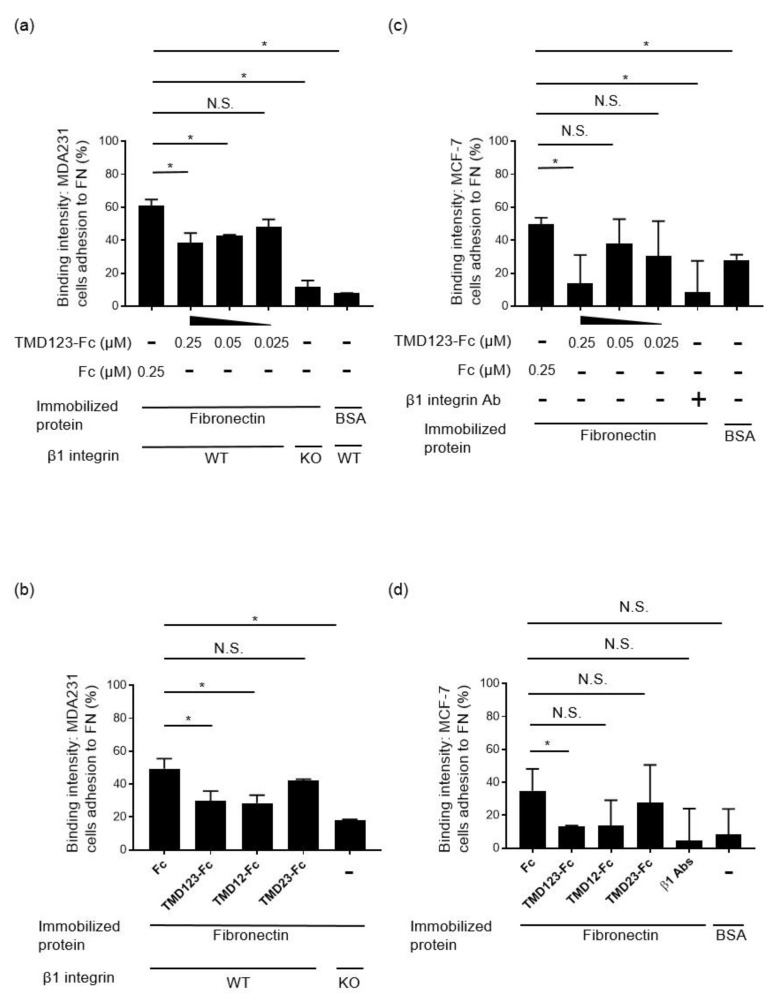
TMD123-Fc and TMD12-Fc inhibit the binding of MDA-MB-231 to fibronectin (FN). (**a**) TMD123-Fc inhibits the binding of fibronectin to MDA-MB-231 in a concentration-dependent manner. (**b**) Lectin-like domain of TM inhibits the β1 integrin-dependent binding of MDA-MB-231 to fibronectin. (**c**) TMD123-Fc inhibits the binding of fibronectin to MCF-7 in a concentration-dependent manner. (**d**) The lectin-like domain of TM inhibits β1 integrin-dependent binding of MCF-7 to fibronectin. V-wells were coated with 10 µg/mL fibronectin or 10 µg/mL BSA and then injected with calcein-stained MDA-MB-231 and subjected to shear stress in the presence of Mg^2+^ and Ca^2+^. Experiments were performed in triplicate and repeated three times with similar results. Data are expressed as mean ± SD. * *p* < 0.05 compared with control. N.S.: not significant; BSA: bovine serum albumin, Ab: antibody.

**Figure 6 biomedicines-09-00162-f006:**
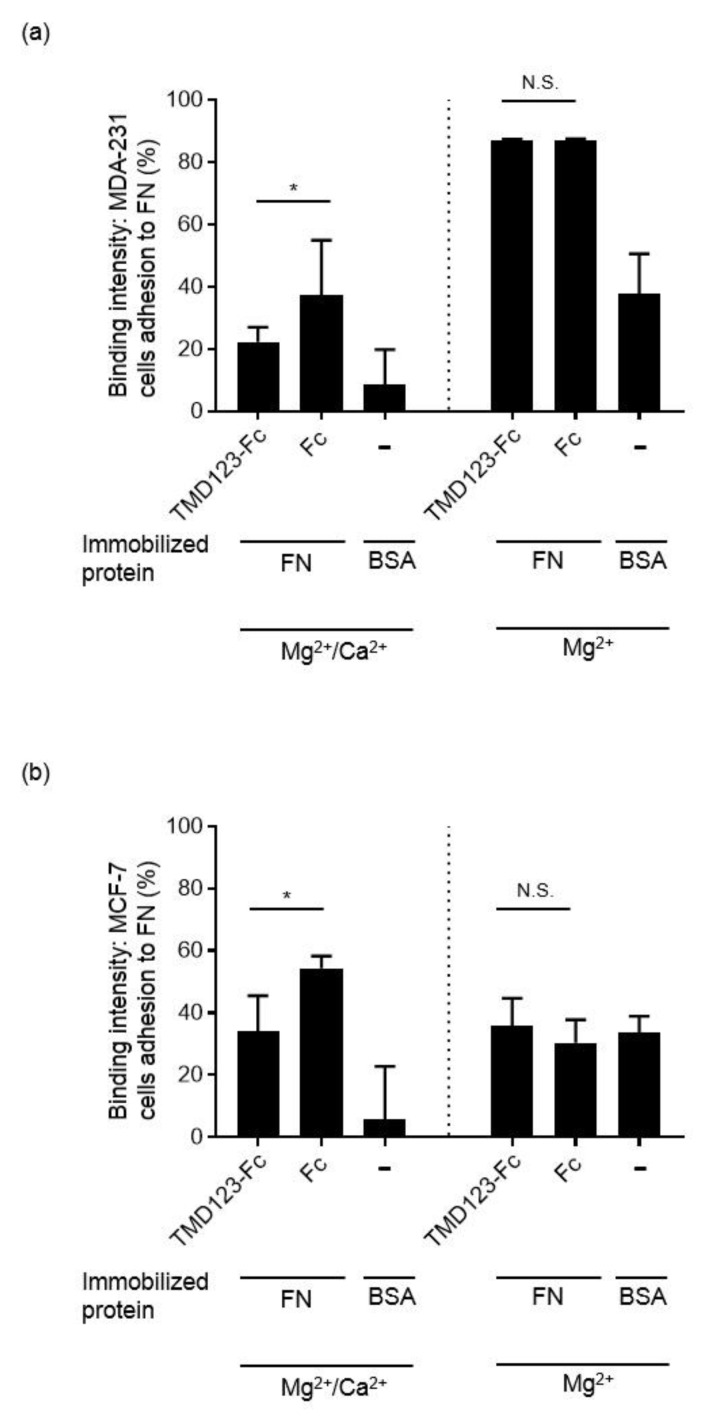
TMD123-Fc inhibits the binding of activated β1 integrin to (**a**) MDA-231 or (**b**) MCF-7 cells and fibronectin (FN) under shear stress. The binding of TMD123-Fc to FN was analyzed by applying shear stress using V-well wells in the presence of 1 mM Mg^2+^ and Ca^2+^ or 1 mM Mg^2+^ alone. Experiments were performed in triplicate and repeated three times with similar results. Data are expressed as mean ± SD. * *p* < 0.05. Compared with control. N.S.: not significant; BSA: bovine serum albumin.

**Figure 7 biomedicines-09-00162-f007:**
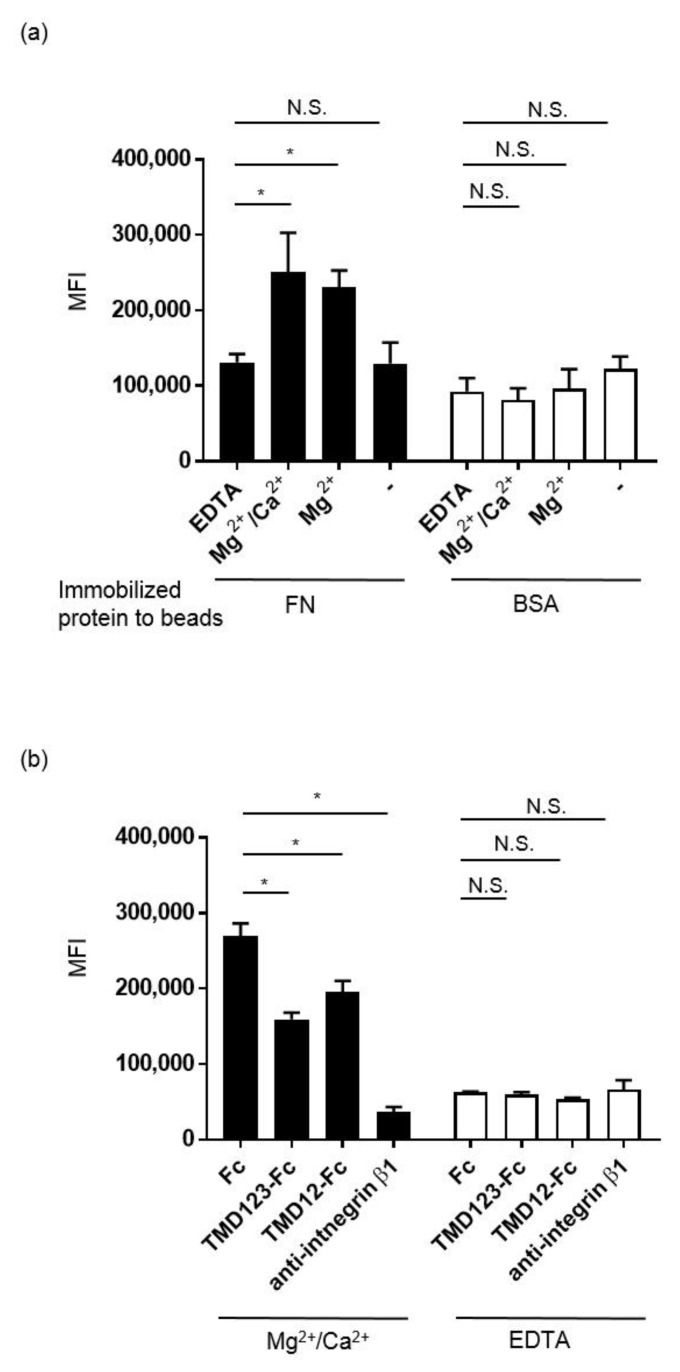
Binding of MDA-231 cells to fibronectin is inhibited by TMD123-Fc. (**a**) Binding of fibronectin- or BSA-coated fluorescent beads (green) to MDA-231 was analyzed by flow cytometry in the presence of 1 mM Mg^2+^/Ca^2+^, 1 mM Mg^2+^, or 2 mM EDTA. (**b**) Binding of fibronectin-coated fluorescent beads (green) to MDA-231 was inhibited by the β1 integrin antibody or TMD123-Fc and TMD12-Fc in the presence of 1 mM Mg^2+^/Ca^2+^. Experiments were performed in triplicate and repeated three times with similar results. Data are expressed as the mean ± SD. * *p* < 0.05. compared with control. N.S.: not significant; MFI: mean fluorescent intensity.

## Data Availability

Please contact to the corresponding author for analyzed data.

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
