# Peer review of "The Lectin-Like Domain of Thrombomodulin Inhibits β1 Integrin-Dependent Binding of Human Breast Cancer-Derived Cell Lines to Fibronectin"

_biomedicines, 2021, doi:10.3390/biomedicines9020162_

Round 1

Reviewer 1 Report

Thank you for considering and implementing the comments from the 1st review. Especially now that the materials and methods section is more extensive, the work is now fine.  The inclusion of the details of the antibodies used (ln 164-187) will be especially valuable for workers wishing to repeat the experiments.

I think that the m/s is now perfectly acceptable for publication, and considerably improved for the additions.

I note however, that only letting the cells attach for 5 min before challenging them with shear stress is not a very hard test.  Perhaps, in a future study, the authors would like to consider how adhesion is affected after longer times of attachment?  I notice that many cell adhesion papers allow cells to attach for 30 min- 1h.

The only thing I note is that "G" is the universal gravitational constant and the authors should use lower case "g" here (ln 113).

Author Response

Responses to Reviewers’ comments

We are grateful for the reviewers’ thoughtful comments and suggestions.  As outlined below, we have addressed and/or clarified each of the issues previously raised in a point-by-point fashion. We believe that these changes have greatly improved the quality and significance of our study, and that the manuscript is now suitable for publication in Biomedicines (please note that corrections/changes throughout the revised manuscript appear highlighted in yellow).

To the Reviewer 1,

C1: The only thing I note is that "G" is the universal gravitational constant and the authors should use lower case "g" here (ln 113).

R1: According to the reviewer’s suggestion, we have revised manuscript (in 113).

Reviewer 2 Report

I have seneral minor commetns

1. p13, Is  Mn2+  right? 

2. Is it cell type secific or common effect? 

Author Response

Responses to Reviewers’ comments

We are grateful for the reviewers’ thoughtful comments and suggestions.  As outlined below, we have addressed and/or clarified each of the issues previously raised in a point-by-point fashion. We believe that these changes have greatly improved the quality and significance of our study, and that the manuscript is now suitable for publication in Biomedicines (please note that corrections/changes throughout the revised manuscript appear highlighted in yellow).

To the Reviewer 2,

C1:  p13, Is  Mn2+  right? 

R1: According to the reviewer’s suggestion, we have revised manuscript (p13, line 361).

C2:  2. Is it cell type secific or common effect? 

R2: Thank you for your question.

Removal of Ca2+ or addition of Mn2+ will strikingly increase ligand binding affinity and adhesiveness of almost all integrins. (ref 1) 

This phenomenon is not specific to MCF-7 or MDA-231 integrins, but is a common phenomenon seen in other cells.

Ref)

  1. The regulation of integrin function by divalent cations. Kun Zhangand JianFeng Chen  Cell Adh Migr. 2012 Jan 1; 6(1): 20–29. PMID: 22647937

This manuscript is a resubmission of an earlier submission. The following is a list of the peer review reports and author responses from that submission.

Round 1

Reviewer 1 Report

I made comments list. please give your response to me.

 Major comments

  1. Put in TM123, TM12 and FN interaction data using integrin KO cell by WB and immunostaining.

       Author already have integrin KO, therefore you can easily confirm this experiment.

  1. Please check integrin expression in KO cell. Author just mentioned knockout results, but need real

  data in this part. It is very important to believe in your result using KO cell.

  1. If possible, check cellular function of TM123, TM 12 in breast cancer cell lines.

Minor Comments

  1. p7, 203 lane, please all change Ca to Ca2+ as well as Mg
  2. p4, lane 155, please put in D1 in lectin like domin and also others….
  3. ex) lectin-like domain (D1)…
  4. Could you change MDA-MB-231 to integrin?

   you have many abbreviations in your manuscript, very confusing.

Author Response

Responses to Reviewers’ comments

We are grateful for the reviewers’ thoughtful comments and suggestions.  As outlined below, we have addressed and/or clarified each of the issues previously raised in a point-by-point fashion. We believe that these changes have greatly improved the quality and significance of our study, and that the manuscript is now suitable for publication in Biomedicines (please note that corrections/changes throughout the revised manuscript appear highlighted in yellow).

To the Reviewer 1,

C1: Put in TM123, TM12 and FN interaction data using integrin KO cell by WB and immunostaining.

 Author already have integrin KO, therefore you can easily confirm this experiment.

R1:  I agree with the reviewer's comment that another experiment should be added that aims to show, in a different setting, that TM inhibits the binding of fibronectin to β1 integrins in MDA-231. As the suggested experimental approach to examine the interaction with TMD123 and FN using integrin KO cell by WB and immunostaining is currently technically diificult for us, we have performed an alternative approach using a flow-cytometry based measurement of FN-coated beads bound to WT and KO cells, as follows.

Fibronectin or BSA was adsorbed onto the surface of fluorescent beads (green; diameter: 3 μm), and binding of MDA-231 to the beads was confirmed by flow cytometory in the presence of 1 mM Mg2+/Ca2+, 1 mM Mg2+, and 2 mM EDTA (Figure 7a). We found that fibronectin-coated beads bound MDA-231 β1 integrin in the presence of 1 mM Mg2+/Ca2+ and 1 mM Mg2+ but not 2 mM EDTA. Furthermore, the binding of fibronectin-coated beads to MDA-231 cells was inhibited by TMD123-Fc and TMD12-Fc (Figure 7b).

We have added the methods (page 5, line 185) and the results (page 12, line 304) of the new experiment, and discuss the results in the Discussion section (page X, line Y) of the revised manuscript.

C2:  Please check integrin expression in KO cell. Author just mentioned knockout results, but need real data in this part. It is very important to believe in your result using KO cell.

R2:  As suggested by the reviewer, we have included the FACS data for MDA-MB-231 (Fig. 3) and β1 integrin-KO cells (Fig. 4) in the revised manuscript. Supplemental Figure 1 shows MCF-7 integrin expression.

C3: If possible, check cellular function of TM123, TM 12 in breast cancer cell lines.

R3: We would like to thank this reviewer for his/her constructive feedback. We will investigate the cellular function of TM in breast cancer cell lines in our future work.

C4: p7, 203 lane, please all change Ca to Ca2+ as well as Mg

R4: Ca and Mg notations in the manuscript have been changed to Ca2+ and Mg2+.

C5: p4, lane 155, please put in D1 in lectin like domin and also others….

  1. ex) lectin-like domain (D1)…

R5: We have added descriptions of the lectin-like domain (domain 1; D1), EGF-like domain (domain 2; D2), and serine-threonine-rich domain (domain 3; D3) to page 5 (Section 3.1) of the revised manuscript.

C6: Could you change MDA-MB-231 to integrin?

R6: Acoording to the reviewer’s suggestion, we have made appropriate changes in the revised manuscript..

C7:   you have many abbreviations in your manuscript, very confusing.

R7: As the reviewer suggested, we have properly defined all of the abbreviations in revised manuscript.

Reviewer 2 Report

Kawamoto et al. examine the extracellular domains of thrombomodulin (TM), a transmembrane protein found on endothelial cells that is an important regulator of thrombin activity and so the coagulation cascade.

They confirm Hsu et al. (Oncotarget, 2016) that TM binds fibronectin (Fig.2), and use integrin b1 knockout (KO) of the cell line MDA-MB-231 to suggest that the lectin-like domain of TM inhibits b1-integrin-dependent binding of breast cancer cells with activated integrins to FN.  As recombinant TMD123 is in clinical usage, they suggest that TM might be a potential adjunct therapy for metastatic breast carcinoma.

I have several issues with the m/s as it stands, many of which would be quite easy for the authors to fix.

Materials and methods:

1) MDA-MB-231: was derived from human breast carcinoma.  It is not a human breast carcinoma. Whether it is representative of breast carcinoma in humans is debatable. So the title is misleading. I suggest " ... binding of a human breast cancer- derived cell line to fibronectin".

The source of the cell line must be given. In view of the reproducibility crisis, it would be valuable if the authors had the identity of the cells they use verified by STR analysis.

2) The m/s would be stronger if the authors had used more than one breast cancer derived cell line in the studies - so that the reader does not think this a peculiarity of this one cell line.

3) §2.3 V-well adhesion assay.  The authors suggest that this assay mimics physiological shear stress to activate integrins. Where is the data to support this suggestion?

While the assay gives an excellent reproducible washing stress (in contrast to pipetting in wash fluid), there is no indication that this is physiological as they claim.  For that the authors would need to go into depth providing local forces (dynes/cm2) acting on the cells in comparison to the flow stresses in blood vessels. The g-force acting on the cells (or even just in the centrifuge) is also not given.  As the original V-adhesion assay papers calibrated these parameters, and showed that they varied between cell lines.  The authors do not show such data for MDA_MB231.  It would be better if they showed  calibration data for the assay, but if they want to avoid this, they must at the very least remove the phrase "physiological shear stress" from the m/s. " Shear stress" would be OK.

  • The authors do not say how long the cells were allowed to attach, and what temperature they were attached at before centrifugation.  These details must please be added.
  • The authors state (%) as the Y-axis parameter.  Could they please define what this is a percentage of?  Maximum cell adhesion? Number of cells added?  Please clarify in materials and methods.

4) §2.7. Concentrations of primary antibodies used, times and temperatures of incubation for flow cytometry must be included.

5) §2.8 catalog numbers of antibodies used are sometimes missing. Please add.  Batch numbers of antibodies used should be included to avoid ambiguity. 

6) §3.1 it would help the reader if the authors would translate μM to μg/ml when referring to coating solutions. Perhaps once in the M&M section would be useful for the reader?

7) Integrins. "b1 integrin" does not exist.  All integrins are obligate heterodimers, and never appear as individual chains at the cell surface, so "b1 integrins" (there are 12 of them...) is correct.

This brings me to a major issue: the use of b1 integrin KO as an indicator of a5b1 involvement in TM/fibronectin interaction. The elderly MDA-MB 231 cell line is subjected to extensive selection pressure following KO of b1 integrin chain by CRISPR.  At least 4 b1 integrins are deleted, as shown by flow cytometry. We do not hear about the other 8 b1 integrins.  

This is a very severe stress for an adherent cell.  In short, given the multiple functions of integrins, I would predict that very many biochemical pathways, and protein expression patterns are perturbed by this KO.

I can accept that the b1 KO provides a hint that a5b1 is involved.  But I am puzzled that the authors did not KO the a5 chain instead.  Surely their argument would be much stronger if they did that? For example in Fig.3 the critical integrin could have been avb1 or a8b1, both of which would have been removed by the b1-KO.  a5b1 may not have been involved.  At least the authors should briefly discuss these possibilities and the weakness of the pan b1 ko.

8) In all figures SD is shown, but number of replicates is missing.  Please include. 

9) §4 Discussion.  Ln 2-3.  Sadly, despite many trials, regulation of integrins is ineffective as a therapeutic strategy for treating cancer (see, for example, Raab et al. 2017).  The authors may therefore want to edit this statement, which is still to my knowledge incorrect. Especially: Voloxicimab, the a5b1 specific therapeutic failed in clinical trials.  Of course, these trials were limited.  But as yet, there is only negative evidence.

1) §4. Para-2. Breast cancer metastasis. Don't breast cancers metastasize predominantly through lymphatic routes?  A therapy targeting an interaction with endothelial lumen, like TM123, may not be very useful.  Plus, the strategy would require a very long half life of the drug to be practical.

Author Response

Responses to Reviewers’ comments

We are grateful for the reviewers’ thoughtful comments and suggestions.  As outlined below, we have addressed and/or clarified each of the issues previously raised in a point-by-point fashion. We believe that these changes have greatly improved the quality and significance of our study, and that the manuscript is now suitable for publication in Biomedicines (please note that corrections/changes throughout the revised manuscript appear highlighted in yellow).

To the Reviewer 2,

C1: MDA-MB-231: was derived from human breast carcinoma.  It is not a human breast carcinoma. Whether it is representative of breast carcinoma in humans is debatable. So the title is misleading. I suggest " ... binding of a human breast cancer- derived cell line to fibronectin".

R1: Acoording to the reviewer’s suggestion, we have changed the title as follows: The lectin-like domain of thrombomodulin inhibits β1 integrin-dependent binding of human breast cancer-derived cell lines to fibronectin.

C2: The source of the cell line must be given. In view of the reproducibility crisis, it would be valuable if the authors had the identity of the cells they use verified by STR analysis.

R2: We agree with the reviewer’s suggestion. In the Materials and Methods, we have described that MDA-231 and MCF-7 cells were purchased from ATCC (page 3, line 129).

C3: The m/s would be stronger if the authors had used more than one breast cancer derived cell line in the studies - so that the reader does not think this a peculiarity of this one cell line.

R3: According to the reviewer's suggestion, we have performed additional experiments using MCF-7 cells. MCF-7 and MDA-231 breast cancer cell lines are representative of luminal-like and mesenchymal-like breast cancer types, respectively. We have shown that the binding of fibronectin to MCF-7 cells was inhibited by TMD123-Fc (Figs. 5c, 5d, and 6b).

C4: §2.3 V-well adhesion assay.  The authors suggest that this assay mimics physiological shear stress to activate integrins. Where is the data to support this suggestion?

While the assay gives an excellent reproducible washing stress (in contrast to pipetting in wash fluid), there is no indication that this is physiological as they claim.  For that the authors would need to go into depth providing local forces (dynes/cm2) acting on the cells in comparison to the flow stresses in blood vessels. The g-force acting on the cells (or even just in the centrifuge) is also not given.  As the original V-adhesion assay papers calibrated these parameters, and showed that they varied between cell lines.  The authors do not show such data for MDA_MB231.  It would be better if they showed  calibration data for the assay, but if they want to avoid this, they must at the very least remove the phrase "physiological shear stress" from the m/s. " Shear stress" would be OK.

R4: We agree with the reviewer’s suggestion. We have changed the phrase "physiological shear stress" to "shear stress" in the revised manuscript.

C5: The authors do not say how long the cells were allowed to attach, and what temperature they were attached at before centrifugation.  These details must please be added.

R5: As the reviewer suggested, we have added details to the Materials and Methods section (page 3, lines 108 and 109) of the revised manuscript as follows:

Before centrifugation, V-wells with cells were incubated at 37°C for 5 min.

C6: The authors state (%) as the Y-axis parameter.  Could they please define what this is a percentage of?  Maximum cell adhesion? Number of cells added?  Please clarify in materials and methods.

R6: We agree with the reviewer’s suggestion. Binding affinity (%) was determined according to the equation described in the manuscript. We have followed the reviewer's suggestion and, thereby, rewritten the Materials and Methods (page 3, lines 113–125) and corrected the Y-axis notation in the figure5 and figure 6.

We calculated the binding affinity between cells and integrin ligands, as follows:

Binding Affinity (%) = {FL(EDTA) – FL(Mg2+/Ca2+)} / FL(EDTA) × 100

where FL(EDTA) represents the fluorescence intensity of integrin binding to the integrin ligand in the presence of 2 mM EDTA, and FL(Mg2+/Ca2+) represents the fluorescence intensity of integrin binding to the integrin ligand in the presence of 1 mM Mg2+/Ca2+. In the presence of 2 mM EDTA, non-adherent cells were concentrated at the bottom of the V-well to elicit an increase in fluorescence intensity. In the presence of 1 mM Mg2+/Ca2+, integrins were activated, and cells adhered to integrin ligands attached to the V-well, resulting in decreases in fluorescence intensity. Introduction of an adhesion inhibitor, such as an integrin antibody or TM, resulted in aggregation of non-adherent cells at the bottom of the V-well and increased fluorescence intensity.

C7:  §2.7. Concentrations of primary antibodies used, times and temperatures of incubation for flow cytometry must be included.

R7: We have included the information requested by the reviewer, thereby modifing the manuscript, as follows (page 4, lines 155–159):

MDA-MB-231 and MCF-7 cells were grown to 90% confluence, detached from the culture dish using trypsin with 1 mM EDTA, washed with FACS buffer (PBS with 2 mM EDTA, 2% BSA, and 0.05% NaN3), and stained with the β1 integrin primary antibody (10 µg/mL) for 30 mins at room temperature. The sample was then incubated with secondary antibody (FITC-anti-mouse IgG, 15 µg/mL) for 30 mins at room temperature.

C8: §2.8 catalog numbers of antibodies used are sometimes missing. Please add.  Batch numbers of antibodies used should be included to avoid ambiguity. 

R8:  We agree with the reviewer’s suggestion. We have included the full manufacturer information for all antibodies in the Material and Methods section (page 4, lines162–183).

C9: §3.1 it would help the reader if the authors would translate μM to μg/ml when referring to coating solutions. Perhaps once in the M&M section would be useful for the reader?

R9: We agree with the reviewer’s suggestion. We have changes the units to ug/ml, as follows (page 3, lines 107 and 108; and page 3, lines 145 and 146).

C10: Integrins. "b1 integrin" does not exist.  All integrins are obligate heterodimers, and never appear as individual chains at the cell surface, so "b1 integrins" (there are 12 of them...) is correct.

This brings me to a major issue: the use of b1 integrin KO as an indicator of a5b1 involvement in TM/fibronectin interaction. The elderly MDA-MB 231 cell line is subjected to extensive selection pressure following KO of b1 integrin chain by CRISPR.  At least 4 b1 integrins are deleted, as shown by flow cytometry. We do not hear about the other 8 b1 integrins. 

This is a very severe stress for an adherent cell.  In short, given the multiple functions of integrins, I would predict that very many biochemical pathways, and protein expression patterns are perturbed by this KO.

I can accept that the b1 KO provides a hint that α5β1 is involved.  But I am puzzled that the authors did not KO the a5 chain instead.  Surely their argument would be much stronger if they did that? For example in Fig.3 the critical integrin could have been avb1 or a8b1, both of which would have been removed by the b1-KO.  a5b1 may not have been involved.  At least the authors should briefly discuss these possibilities and the weakness of the pan b1 ko.

R10: 

We would like to thank this reviewer for asking an important question about a potential off-target effect induced by the CRISPR-KO experiment procedures including transfection and puromycin selection. In addition we acknowledge the reviewer’s comment that multiple b1 integrins might be involved in the interactions observed in this study.  We have discussed these points in the Discussion section of the revised manuscript (page 14 line 376-).

C11: In all figures SD is shown, but number of replicates is missing.  Please include. 

R11:  We agree with the reviewer’s suggestion. All experiments were repeated at least three times. This statement was added to the figure legends.

C12: 9) §4 Discussion.  Ln 2-3.  Sadly, despite many trials, regulation of integrins is ineffective as a therapeutic strategy for treating cancer (see, for example, Raab et al. 2017).  The authors may therefore want to edit this statement, which is still to my knowledge incorrect. Especially: Voloxicimab, the a5b1 specific therapeutic failed in clinical trials.  Of course, these trials were limited.  But as yet, there is only negative evidence.

R12: We agree with the reviewer’s comment.  We have modified the Discussion section as follows (page 13, lines 323–329):

The integrin family of cell-adhesion receptors regulates a diverse range of cellular functions important for adhesion, migration, and metastasis of solid tumors. Integrin α5β1 recognizes and attaches to extracellular ligands containing RGD tripeptide motifs. Aberrant upregulation of integrin α5β1 has been implicated in many human malignancies and is closely associated with poor prognosis [22]. Therefore, studies on the molecular interactions between integrin β1 and its ligands are essential for interpreting the biological functions and underlying mechanisms associated with β1 integrin.

C13: 1) §4. Para-2. Breast cancer metastasis. Don't breast cancers metastasize predominantly through lymphatic routes?  A therapy targeting an interaction with endothelial lumen, like TM123, may not be very useful.  Plus, the strategy would require a very long half life of the drug to be practical.

R13:  We agree with the reviewer’s suggestion in that breast cancers metastasize predominantly through lymphatic routes.  We added the following sentences in discussion part (page 14, lines 384-):

Finally, breast cancer most commonly spread via lymphatics[33]. However breast cancer migrates to distant organs by lymphogenous and hematogenous metastasis[34] [35]. Because fibronectin is present in both vascular endothelial cells, lymphatic vessels[36] and lymph nodes[37]. TMD123-Fc might contribute to the inhibition of β1 integrin binding to fibronectin at both vascular endothelial cells and lymph nodes.

We agree with the reviewer’s suggestion in that the strategy would require a very long half life of the drug to be practical.  We added the following sentences in discussion part (page 14, lines 381-):

Second, we found that TMD123-Fc inhibited the binding of human breast cancer-derived cell lines to fibronectin in vitro. However we have not conducted in vivo experiments. A longer half-life may be necessary to prove the efficacy of TMD123-Fc in vivo.
